# Experimental and Numerical Study on Friction and Wear Performance of Hot Extrusion Die Materials

**DOI:** 10.3390/ma15051798

**Published:** 2022-02-27

**Authors:** Leilei Zhao, Kecheng Zhou, Ding Tang, Huamiao Wang, Dayong Li, Yinghong Peng

**Affiliations:** School of Mechanical Engineering, Shanghai Jiao Tong University, Shanghai 200240, China; zhaoleilei@sjtu.edu.cn (L.Z.); kecheng@sjtu.edu.cn (K.Z.); wanghm02@sjtu.edu.cn (H.W.); dyli@sjtu.edu.cn (D.L.); yhpeng@sjtu.edu.cn (Y.P.)

**Keywords:** extrusion, friction, CVD, cemented carbide, FEM

## Abstract

For the aluminium alloys produced by the hot extrusion process, the profile is shaped according to the bearing at the exit of the extrusion die. The tribological process has significant effects on the die service life, profile dimensional tolerances, and profile surface finish. Recently, new technologies have been introduced to the hot extrusion die, such as cemented carbide insert die and surface coating. However, under hot extrusion working conditions, quantitative studies on their friction and wear performances are lacking. In this work, the friction and wear performances of three typical extrusion die materials, traditional hot tool steel (H13), cemented carbide (YG8), and chemical vapour deposition (CVD) coating, were studied. Macro and nano hardness tests, Pin-on-disk friction and wear tests, optical profiler and SEM observations, and experiments and simulations of hot extrusion were conducted. The results show that the coefficients of friction of CVD coatings and H13 hot work tool steel specimens were smaller under the hot extrusion condition than at room temperature. The wear mechanisms of H13, YG8, and CVD coatings at 500 °C are adhesion, abrasive, and fatigue, respectively. Moreover, the tribology results were validated by the extrusion experiments and the finite element analysis of hot extrusion. The conclusion of this manuscript is useful not only for the numerical simulation of the hot extrusion process but also for the surface finishing of the extrusion profile.

## 1. Introduction

The use of precision aluminium profiles is increasing in many industrial equipment, instruments, and electronic products that require assembly mating, light weight, or heat transfer enhancement. Precision aluminium parts usually have the characteristics of tight tolerance of dimensions, complexity, and thin-walls [1,2,3]. These characteristics pose challenges to extrusion dies, especially their service life. A comparison of the cross-section shape and surface of profiles of the extrusion die of a micro channel tube between the initial and later phases of extrusion is shown in Figure 1. The profiles show that wear of the extrusion dies increased the surface roughness and induced size deviation on the tube section. The characteristics of the extrusion die friction and wear are important for precision aluminium extrusion [4,5].

In aluminium extrusion, the profile is shaped according to the bearing (or working-belt) at the exit of the extrusion die, where the exit speed of the profile varies from 0.1 m/s to 1.7 m/s, the die bearing surface temperature can reach 550 °C, and the shear stress on bearing surfaces is usually less than 30 MPa [5,6,7,8,9,10]. For precision extrusion, the extrusion dies are generally made of hot work tool steels, which are nearly solely utilized for extrusion dies. Hot work tool steels have a high degree of strength and ductility, a strong resistance to tempering, and a low cost. Nowadays, cemented carbide insert die and chemical vapour deposition (CVD) coatings are successfully employed to reduce wear. Figure 2 shows typical aluminium extrusion dies with hot tool steel, cemented carbide insert die, and CVD coating. Cemented carbide insert die is more wear-resistant, but it is much more brittle and can break easily. To counter this, CVD coatings are suitable for tool steel, combining the advantages of wear resistance and ductility of the base material. In practice, knowledge of friction and wear characteristics of the materials are required, especially qualitative assessments of the extrusion process conditions.

Progress in bearing surface wear research is critical for establishing adequate tribological conditions and achieving the required surface quality of extrudates. Numerous investigations on the friction and wear properties of cemented carbide and CVD coatings for mold and tool applications have been done [11,12,13,14,15,16]. Khatter et al. [17] investigated the cause of failure in cemented carbide extrusion dies and indicated that the failure occurred as a result of aggressive cemented carbide machining. In addition, according to experiments under different machining conditions on cemented carbide materials, the optimal machining settings have been found, and a new die has been developed and manufactured, which exhibits high performance. List et al. [13] investigated the wear behavior of cemented carbide tools during dry machining of an aluminum alloy with a cemented carbide insert and discovered that adhesion and diffusion play a role in the wear process. Coating technologies have been widely studied and applied in other mold fields [18,19,20,21,22]. For example, Wang et al. [21] studied the effects of ceramic PVD coatings on the heat of steel, investigating the resistance and aluminium corrosion resistance, as well as changes in hardness and impact toughness after thermal cycling fatigue tests. Mitterer et al. [23] presented and discussed results obtained on hard coatings applied to die-casting dies using sputtering and plasma-assisted chemical vapour deposition (PACVD) as well as its application in aluminium die casting. Currently, coatings for extrusion dies created using chemical vapour deposition (CVD) are often multi-layer systems, which combine the advantages of TiC, TiN, and TiCN single-layer coating, with advantages of high hardness, low coefficient of friction (COF), corrosion resistance, and good thermal stability [24,25,26,27,28,29,30].

The wear performance of the extrusion bearing has been studied experimentally. In a system that simulates bearing wear in an extrusion die, Bjork et al. [31] evaluated commercial and experimental physical vapour deposition (PVD) coatings. Muller [32] used the Knoop-hardness test and the pin on disk friction test to investigate a coated extrusion die with duplex layers. The critical load was determined via a scratch test. Karadogan et al. [33] studied the friction behaviour in extrusion processes by designing and manufacturing a new cone-friction test for the purpose of analyzing friction phenomena, in which process parameters such as temperature, pressure, and relative velocity can be controlled. Maier et al. [34] investigated various coating designs to verify the advantages of coated dies in aluminium extrusion production. However, typical extrusion die materials have not been compared under the same testing standard of friction and wear thus far.

In this work, the tribological characteristics of the hot tool steel cemented carbide and CVD coating were comprehensively investigated. The macro and nano hardness of three types of the specimen were tested, and standard pin-on-disk friction tests under both room temperature and temperature of hot extrusion were then carried out. The cross-section profiles of the wear scars and wear rates were examined using a non-contact surface profilometer. Finally, the wear tracks and wear mechanisms of the materials were observed through SEM, and the tribology results were validated by the finite element method of multi-hole flat-tube extrusion.

## 2. Experiment Procedure

### 2.1. Materials and Specimens

Three different materials, namely H13 hot work tool steel, cemented carbide YG8, and CVD coating, were used in friction and wear experiments. The chemical compositions of the studied die materials and the counter friction specimen are shown in Table 1. Specimens were fabricated as flat disks with a diameter of 30 mm and a thickness of 5 mm. The counter friction specimen was a flat-ended cylindrical pin (∅5 mm and 16 mm long) made from cemented carbide YG6. The specimen of H13 hot work tool steel with coating was prepared via chemical vapour deposition by Kunshan Socket Surface Technology Co., Ltd. (KunShan, China) Before deposition, the H13 disk, and sputtering target were cleaned for a further 1200 s through argon ion beam sputtering generated at energies ranging from 1500 to 2000 eV and regulated by the plasma voltage. The pressure of deposition was 0.8 Pa. The main components of the coating are TiCN, TiN, and Al_2_O_3_. Temperature, pressure, gas composition, and flow rate all affect the coating process [34].

In order to compare the hardness of the specimens, both macro and nano hardness tests were carried out. For the CVD coating, the observation position is shown in Figure 3a. Macro hardness was tested with a Metallic Rockwell Hardness Tester HR-150DT (Suiou, Shanghai, China), and nano hardness, which shows the hardness distribution of the multilayer coating, was tested using the Nanomechanical Test Instrument Hysitron Ti 950 (Figure 3b) by Hysitron Co., Ltd. equipped with a Berkovich diamond indenter (Bruker, Billerica, MA, USA). The indenter has a tip radius of 50 nm, angle of 140.6°, dimensionless correlation factor of 1.034, Poisson’s ratio of 0.07, and Young’s modulus of 1141 GPa. The nanoindentation position of the CVD coating is shown in Figure 3a. For the CVD coating, force-controlled indentation measurements were conducted at the speed of 10 nm/s up to 14,000 μN. For the YG8 and H13, displacement-controlled indentation measurements were conducted at the speed of 10 nm/s up to 500 nm. The load-displacement curve for each nanoindentation is illustrated in Figure 3d, from where we can find that the coating modulus of the surface layer is the largest, followed by the middle layer, and the innermost layer is the smallest. The results of macro hardness and nano hardness measurements show that the nano hardness of multilayer coating decreases gradually from outside to inside, and the outermost layer nano hardness of the multilayer coating is quite high compared to the cemented carbide YG8 and H13 hot work tool steel (Table 2). The hardness of the relative standard deviations is less than 5%. This suggests that the CVD coated die has a higher wear resistance than dies made of H13 hot work tool steel or cemented carbide [26,27]. Moreover, because the nano and macro hardness of the cemented carbide YG6 are higher than that of other tested materials, the cemented carbide YG6 is suitable for the counter specimen.

### 2.2. Test Equipment

The friction and wear tester employed in this investigation was a pin-on-disk (PoD) friction and wear tester, as seen in Figure 4. The friction and wear tester’s electromagnetic motor enables the lower specimens (tool steel disks) to rotate under normal stress in opposition to a stationary upper test specimen. Standard weights were used to apply the appropriate normal load. A heater was installed on the inside wall of the incubator, which heated the test disks to the desired temperature. The friction and wear tribometer was equipped with a computerized data acquisition and control system that allows for the control and monitoring of the applied load, temperature, friction radius, and motor speed.

### 2.3. Test Procedures

The test parameters were based on normal industrial hot aluminium extrusion. The following are the parameters: a load of 15 N, a nominal pressure of 7.64 MPa, a friction radius of 5 mm, room temperature and 500 °C, a motor speed of 380 r/min, and a corresponding sliding velocity of 0.0317 m/s, which is close to the extrusion speed [35]. The test time is 1800 s, corresponding to sliding distances of 358 m.

All disks and pins were ultrasonically cleaned in ethanol for 300 s prior to testing. Firstly, the thermocouple heated the incubator to the desired temperature. During the heating process, the lower specimen (the disk) was rotating at a set rotation speed, and the cooling water system of the friction and wear tester was opened to protect the machine from overheating. Additionally, during the heating phase, the upper pin specimen was kept separate from the disk. After the disk achieved the test temperature, the pin was brought into contact with it, followed by the application of the load and the performance of the test. After the tests, each specimen was cleaned ultrasonically in ethanol for 300 s. Then, the disks’ wear volume was determined using the Zegage Optical Surface Profiler produced by Zygo (Tucson, AZ, USA) and scanning electron microscopy (JSM-7800, JEOL, Tokyo, Japan), respectively.

## 3. Extrusion Experiment and Simulation

### 3.1. Extrusion Experiment

In order to validate the results obtained from the friction and wear test, the extrusion experiments and the finite element simulation of the multi-port flat tubes were conducted.

The die holder equipped with six identical porthole dies for extruding multi-port flat tubes is illustrated in Figure 5a,b. Experiments on the multi-hole extrusion method were conducted on a 3500-ton direct extrusion press with billets made of the AA3003 aluminum alloy. The process parameters employed in the extrusion experiment are summarized in Table 3. A pressure sensor was used to determine the maximum extrusion force during an extrusion cycle. Six multi-port flat tubes at the die exit during extrusion are depicted in Figure 5c, and the multi-port flat tube’s cross-section is depicted in Figure 5f.

### 3.2. Finite Element Simulation

The geometric model of the porthole die was built for the FE model. The upper die and lower die portions of the model were extracted, as illustrated in Figure 5d,e, respectively. As shown in Figure 6a, the FE model for multi-port flat-tube extrusion was created using Qform-Extrusion software (QuantorForm, Moscow, Russia) utilizing the Lagrange-Euler technique to avoid mesh distortion [36]. Because the bearing and profile regions will be subjected to high shear deformation, these portions were meshed using extremely fine tri-prism elements. And, as shown in Figure 6b, the element size in the profile region is selected to have at least two layers in the cross-section. The porthole, welding chamber, and billet meshes were all constructed using tetrahedral elements. Aluminium alloy 3003 (AA3003) (Nanping Aluminum, Nanping, China) and H13 hot work tool steels (Zhuzhou precision drilling cemented carbide, Zhuzhou, China) are used for the extrusion billet and dies, respectively, and their physical parameters are described in Table 4. It was assumed that the billet was viscoplastic, and the tool was rigid.

The critical process parameters employed in this simulation were established using real production conditions, as indicated in Table 3. The cylindrical AA3003 billet has a diameter of 228 mm and a length of 1100 mm. The predicted extrusion ratio is around 448.99, which is within the range of a reasonably high extrusion ratio. Due to the thin-walled feature and high extrusion ratio, severe plastic deformation occurs during the extrusion process, which generates a considerable quantity of heat and causes the temperature to rise, resulting in over burning and other surface defects to the extrudate. To avoid this problem, the initial temperatures of the billet and tools were set to 500 and 480 °C, respectively, and the ram speed was set to 3 mm/s. Between the die and billet, the heat convection coefficient is considered to be 3000 W/m^2^ °C. According to the friction and wear test, the relevant interface friction boundary conditions between billet and tools were established.

## 4. Results and Discussion

### 4.1. Coefficient of Friction

Results of the pin-on-disc friction tests on the three different samples under two different temperatures are shown in Figure 7. All three samples entered a relatively stable state after a short run-in stage. The variation amplitude of the curves at 500 °C was larger than that at room temperature, which indicates that the contact condition of the friction pair degrades at high temperature. The average values of the COF are listed in Figure 8 and the scatter bars mean the standard deviation, which shows that cemented carbide YG8 and H13 hot work tool steel exhibit the minimum COFs at room temperature and 500 °C, respectively.

The COF of cemented carbide YG8 and H13 hot work tool steel increased and decreased, respectively, with increasing temperature. SEM images of the samples at high temperature reveal that after the PoD tests, the oxide film was attached to the surface of H13 hot work tool steel (see Figure 9a,b) while the surface of cemented carbide YG8 didn’t show any oxide film (see Figure 9c,d). Velasco et al. [36] observed that tribo-oxidation processes became more prominent as the test temperature increased. The tribo-oxide films could separate the contacting surfaces, decrease the real contact area, and act as lubricants. At elevated temperatures, however, no oxide layer formed on the surface of the CVD coating; instead, a tiny number of oxide particles were seen on the coating’s surface. Because of the existence of oxide particles and the high surface hardness of the CVD coating, the friction type between the friction bolt and the coating surface probably was sliding friction. Therefore, the COF of the CVD coating at high temperature was lower than that at room temperature. Although the COF of H13 hot work tool steel at 500 °C was smaller than that of the CVD coating, the wear rate of H13 hot work tool steel was too high to enhance the service life of the extrusion die (Figure 10).

### 4.2. Wear Rate

A non-contact surface profilometer (Zegage Optical Surface Profiler, Zygo, Tucson, AZ, USA) was adopted to generate high-resolution images and cross-section profiles of the wear scars.

The wear track profiles on the disc surface after pin-on-disc tests at room temperature and 500 °C are shown in Figure 11 and Figure 12, respectively. The volume loss can be obtained according to the cross-section profiles and friction radius. Then, the wear rate of the samples can be calculated by Equation (1) [37]. The wear rate values of the three samples are presented in Figure 10, and the scatter bars are the standard deviation. The best wear resistance material at room temperature and 500 °C are cemented carbide YG8 and CVD coating, respectively. The wear rate at 500 °C is higher than that at room temperature for the three samples. The deterioration of the surface contact state of friction pairs at high temperatures may be responsible for this phenomenon:(1)$$W_{a}=\frac{V}{F_{N}\cdot s}$$
where *V* (mm^3^) denotes the volume loss, *F_N_* denotes the applied normal load, and *s* (m) denotes the sliding distance.

At 500 °C, H13 hot work tool steel exhibited decreased wear resistance. This could be because the wear resistance of H13 hot work tool steel is proportional to its fracture resistance, and cracks would easily form and propagate alongside carbide particles during sliding, speeding fracture and resulting in a higher wear rate [38,39]. Additionally, friction heat causes a rapid rise in temperature on the worn surface at high temperatures, weakening the H13 hot work tool steel behind the oxide layer. Moreover, the hardness of H13 is the smallest among the three die materials (see Table 2). Thus, the underlying H13 hot work tool steel was unable to maintain the oxide layer, accelerating its delamination and resulting in an increased wear rate of H13 hot work tool steel [40]. On the contrary, the multi-layer CVD coating exhibited excellent wear resistance at high temperatures and a smaller difference in wear resistance between high and room temperatures. This can be mainly attributed to the combination of different layer systems in the multi-layer CVD coating that the hardness of the outermost surface of the multi-layer CVD coating system is the highest (see Figure 3c), providing the best results in wear ratio, bond, hardness, and thermal stability [24,25]. We noticed that the wear rate was always higher at an elevated temperature, while this was not always the case for friction. This is because the oxidation wear is intensified at high temperatures, which makes the wear rate at a high temperature higher than that at room temperature. Under the condition of dry friction, the oxide at the friction interface has a decisive influence on the friction properties. On the one hand, the oxide film plays a lubricating role and reduces the COF. On the other hand, the oxide film is prone to fatigue spalling, resulting in the increase of COF. At high temperatures, a thick oxide film is formed on the surface of the H13 sample, which reduces the COF. The CVD coating has a smooth surface due to its high surface hardness. YG8 is not easy to form a stable oxide film on the surface due to its brittleness, so the surface is rough at high temperatures, which increases the COF.

### 4.3. Effect of Coefficient of Friction on Extrusion Load

To verify the reliability of the COF determined through the friction and wear test, we set the relevant COF at 500 °C (see Figure 8) in the finite element of microchannel tube extrusion simulation to get the simulation results of maximum extrusion load (see Figure 13). As the extrusion process of microchannel tubes occurs in a high pressure and vacuum environment, it is hard to test the parameters related to the friction properties of the extrusion die. The maximum extrusion load is primarily determined by the COF between the billet and the extrusion die, so we compare the maximum extrusion load of the test and simulation to evaluate the rationality of the COF in the simulation process. As we all know, the larger the COF between the die and the billet, the greater the necessary extrusion force. In our study, the order of maximum extrusion force of different types of dies obtained from FE simulation is consistent with the order of COF measured by friction and wear test that the maximum COF is the cemented carbide YG8, followed by coating die, and the H13 hot work tool steel is the smallest as shown in Figure 8. The relative order of maximum extrusion forces for different types of dies obtained from FE simulation and experimental extrusion (Figure 13) is consistent with that of COFs. Besides, the peak extrusion force obtained by simulations is in good agreement with that obtained by extrusion experiments. In the actual extrusion production of a microchannel flat tube, the maximum power of the extruder is only allowed to be 80% of the rated power of the extruder to protect the extruder because the maximum extrusion force will fluctuate by about 10% during the operation of the extruder. Therefore, in this paper, 10% scatter bar was shown in the test data. The difference between the test and simulation is within the allowable error range. Therefore, through the above analysis, it can be concluded that the extrusion finite element model’s boundary conditions obtained by friction and wear test are reliable, which establishes that the COFs determined by friction and wear tests are reasonable.

### 4.4. Wear Mechanism

A series of scanning electron microscope (SEM) observations and energy dispersive spectroscopy (EDS) tests were done to better understand the wear mechanism of H13 hot work tool steel, cemented carbide YG8, and CVD coating under various test settings.

SEM images of the worn surfaces of H13 hot work tool steel subjected to a 15 N load at room temperature are shown in Figure 9a,b. On the worn surface, a considerable number of particles of various sizes were discovered, and the worn surface of H13 hot work tool steel also exhibited small flaking pits and mild micro-grooves. As the metal was detached from the surface of the sample, it rapidly oxidized to form hard oxide particles. Because the hardness of oxide particles is much higher than that of H13 hot work tool steel, under the pressure of the counter bolt, the oxide particles can penetrate the surface of the hot tool steel sample and produce plastic deformation to form grooves and pits when the counter bolt and the sample slide relatively, which is consistent with the characteristics of abrasive wear. Thus, the wear mechanism of H13 hot work tool steel at room temperature is predominantly abrasive wear. The worn surface of cemented carbide YG8 (Figure 9c,d) also shows some particles with different sizes, craters, and slight grooves. Therefore, the wear mechanism of cemented carbide YG8 is also governed by abrasive wear at room temperature.

The CVD coating’s wear track surface morphology exhibits slight delamination and a small amount of wear debris, as shown in Figure 9e,f. In addition, the wear scars are characterized by flaking pits. The oxide particles cannot penetrate the CVD coating to produce plastic deformation as their hardness is lower than that of the coating, and with high hardness, the CVD coating is unlikely to fall off during the test. Besides, EDS analysis (Figure 14c) show no Fe element exists on the scanning surface, which confirms that the CVD coating structure is still intact. This is fairly consistent with the characteristics of fatigue wear, which is a process of material removal, such as pitting, shallow or deep peeling, on the contact surface after a certain number of cycles. Therefore, for the case of CVD coating at room temperature, the wear mechanism is fatigue wear.

SEM images of the worn surface of H13 hot work tool steel after sliding at 500 °C are shown in Figure 15a,b, which clearly show that a discontinuous oxide film adhered to the sample surface, and slight grooves also developed on the worn surface. This suggests that tearing and detachment occurred during the wear process. This may be because the surface of the H13 sample was softened, and the hardness decreased with increased temperature. Therefore, under the same pressure, the friction bolt was pressed into the H13 sample deeper than that at room temperature. Under the combined action of load and motor drive, materials were easily taken from the surface of the H13 sample by the friction bolt, and the detached material readily formed an oxide film on the sample’s surface. The oxide film adsorbed on the sample’s surface acts as a lubricant, as indicated by the average COFs at room and elevated temperatures; the COF at elevated temperatures is lower (Figure 7). Besides, from Figure 16, EDS analysis shows that the content of oxygen is higher at high temperature than at low temperature, which indicates that the surface oxidation of H13 hot work tool steel is more severe at high temperature than at room temperature. These phenomena can be explained by the H13 specimen’s high wear rate at high temperatures (Figure 10). According to the data above, adhesive wear is the predominant mode of wear for H13 hot work tool steel at 500 °C.

Figure 15c clearly shows the appearance of chrysanthemum shapes on the worn surface of cemented carbide YG8 and dendritic cracks in the corresponding magnified positions (Figure 15d), which means that the contact condition of the friction pair degrades at high temperature. The COF of cemented carbide YG8 (Figure 7), being the largest at high temperature, also supports this finding. Slight grooves and a small quantity of debris also appeared on the worn surface. Although cemented carbide YG8 has high hardness and good wear resistance, it has low toughness, and when subjected to impact load at high temperature, it is prone to brittle fracture. As the friction and wear test involved high-speed relative sliding between the friction bolt and the specimen, slight vibration inevitably existed during the test process. This phenomenon is reasonable because slight vibration also occurs during the actual extrusion process. Under the action of slight vibration, the friction bolt will produce an impact load on the sample surface, leading to brittle fracture on the surface of cemented carbide YG8, producing chrysanthemum shapes. Under the combined action of the load and motor driving, the debris produced by brittle fracture of the sample formed grooves on the surface of the sample. Therefore, based on the above analysis, the chrysanthemum shapes on the surface of the specimen are independent of the wear mechanism, and the wear process of cemented carbide YG8 at 500 °C is governed by abrasive wear mechanism.

Figure 15e shows worn surface micrographs of the CVD coating after the PoD test at 500 °C. The surface of the CVD coating sample exhibited barely any wear damage and was much smoother than the worn surface of the H13 hot work tool steel and cemented carbide YG8. Only a few debris and pits were observed randomly dispersed over the surface. In the multilayer coating structure, the hardest coating is the outermost layer, and the microhardness of multilayer coating decreases gradually from outside to inside. One of the benefits of this structure is that the hardest outermost coating provides excellent wear resistance for the specimen. The worn surface micrographs of the CVD coating show little wear damage indicating that the CVD coating hardness plays a key factor in abrasion resistance when the CVD coating has a comparable or relatively higher hardness than the abrasive particles.

Additionally, this structure incorporates a transition between the hardness of the outermost coating and the matrix, which improves the coating’s adhesion to the matrix. The coated sample’s scanning electron micrograph and EDS analysis revealed that the coating did not fall off the substrate, showing that the coating and the substrate have good adhesion. Due to the great hardness and stickiness of coated samples, the probability of abrasive and adhesive wear was fairly low. The pitting corrosion on the coating surface may be caused by fatigue wear. Therefore, the wear mechanism of CVD coating at 500 °C is fatigue wear.

## 5. Conclusions

In this study, the tribological behaviours of H13 hot work tool steel, cemented carbide YG8, and CVD coating sliding against cemented carbide YG6 at different testing temperatures were investigated. The testing conditions closely resembled the normal conditions of the hot extrusion process. Besides, the extrusion experiments and finite element simulation of microchannel tube extrusion are carried out to validate the reliability of the results of tribology tests. The following conclusions can be drawn:(1)The H13 hot work tool steel sample exhibited the lowest hardness among the three tested materials, which further weakened due to thermal softening. It experienced the most extensive wear damage, showing that the traditional hot extrusion dies material needs to be improved.(2)The COFs of CVD coatings and H13 hot work tool steel specimens were smaller under the hot extrusion condition than at room temperature. As the extrusion experiment validates the friction boundary conditions of the FE model, the results of the tribology test are suitable for the numerical simulation of hot extrusion.(3)Due to the fact that cemented carbide YG8 exhibits the best wear resistance and friction reduction at room temperature, and CVD coating exhibits the best wear resistance at high temperature, cemented carbide YG8 is more suitable for extrusion dies working at room temperature, while CVD coating can be used to extend the service life of extrusion dies working at a high temperature.(4)The wear mechanisms of H13 hot work tool steel, cemented carbide YG8, and CVD coatings at 500 °C are adhesion wear, abrasive wear, and fatigue wear, respectively.

## Figures and Tables

**Figure 1 materials-15-01798-f001:**
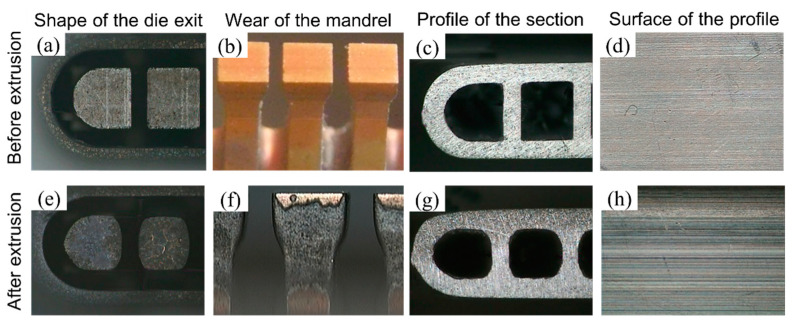
Comparison of microchannel die and products before and after wearing: (**a**) Shape of the die exit before extrusion; (**b**) Wear of the mandrel before extrusion; (**c**) Profile of the section before extrusion; (**d**) Surface of the profile before extrusion; (**e**) Shape of the die exit after extrusion; (**f**) Wear of the mandrel after extrusion; (**g**) Profile of the section after extrusion; (**h**) Surface of the profile after extrusion.

**Figure 2 materials-15-01798-f002:**
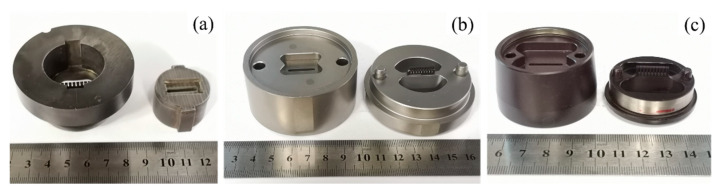
Typical aluminum extrusion die: (**a**) hot tool steel; (**b**) cemented carbide insert die; (**c**) CVD coating.

**Figure 3 materials-15-01798-f003:**
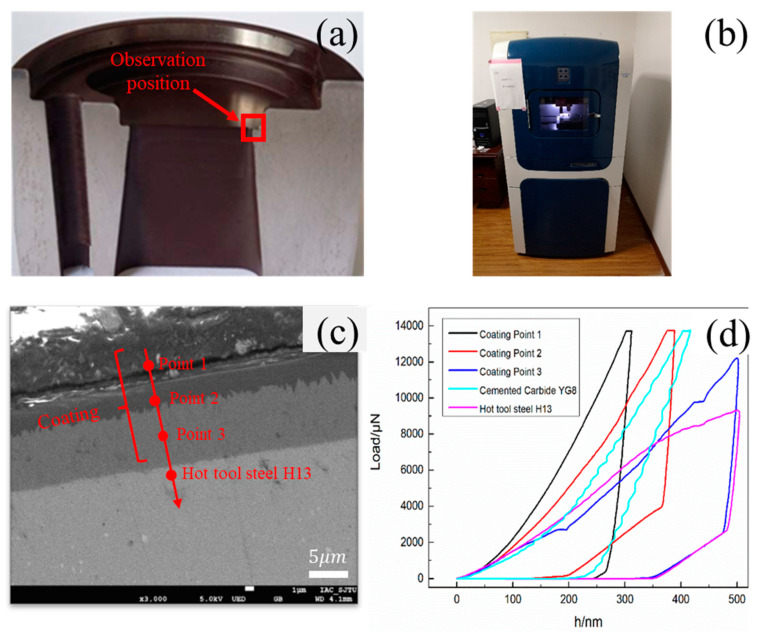
Nanoindentation tests: (**a**) Observation position; (**b**) Nanoindentation equipment; (**c**) Cross-section morphology of the coating; (**d**) Load-displacement curve in nanoindentation hardness measurements.

**Figure 4 materials-15-01798-f004:**
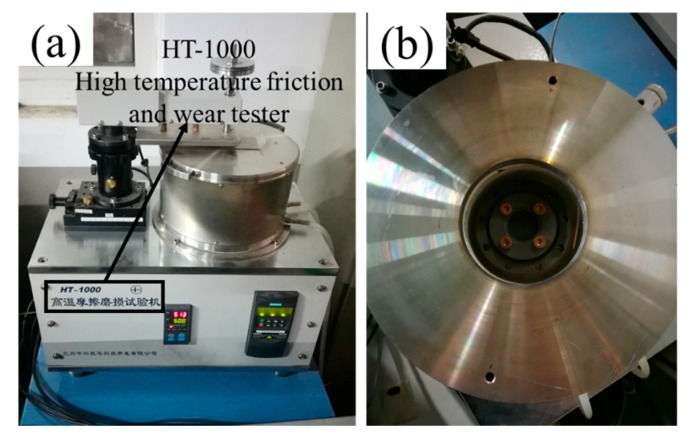
Pin-on-disk friction and wear test equipment: (**a**) testing device; (**b**) specimen heater and fixer.

**Figure 5 materials-15-01798-f005:**
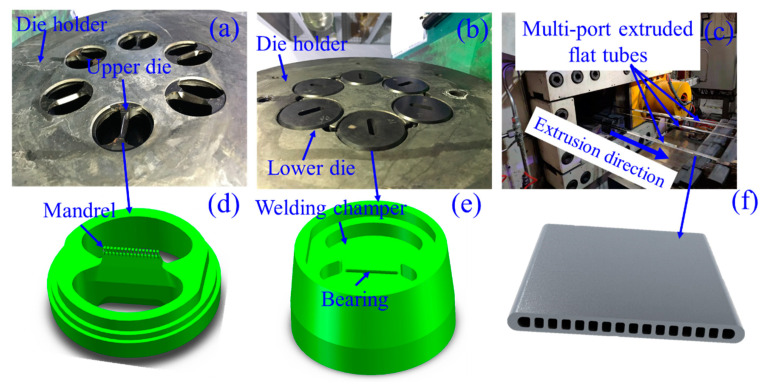
Extrusion experiments and porthole die simulation model (**a**,**b**) die holder with six dies, (**c**) the six multi-port extruded flat tubes at the die exit, (**d**) upper die, (**e**) lower die, (**f**) cross-section of multi-port extruded flat tubes.

**Figure 6 materials-15-01798-f006:**
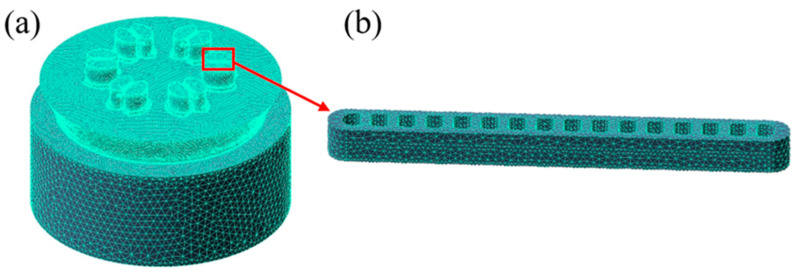
Six-hole extrusion FE model: (**a**) material flow domain and (**b**) profile detail.

**Figure 7 materials-15-01798-f007:**
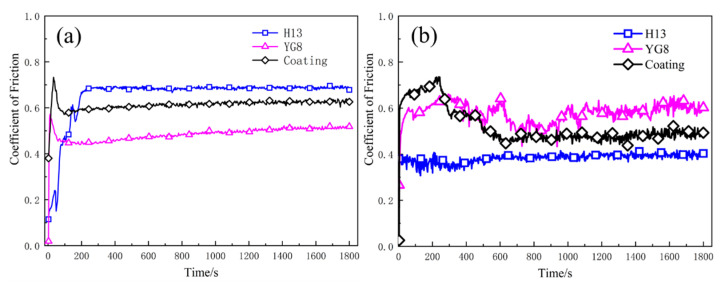
COF of the tests performed at (**a**) 30 °C; (**b**) 500 °C.

**Figure 8 materials-15-01798-f008:**
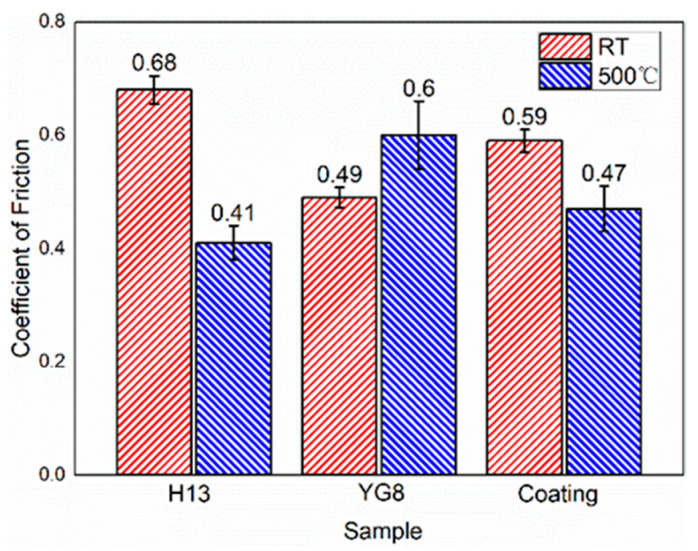
Average COF of different test conditions.

**Figure 9 materials-15-01798-f009:**
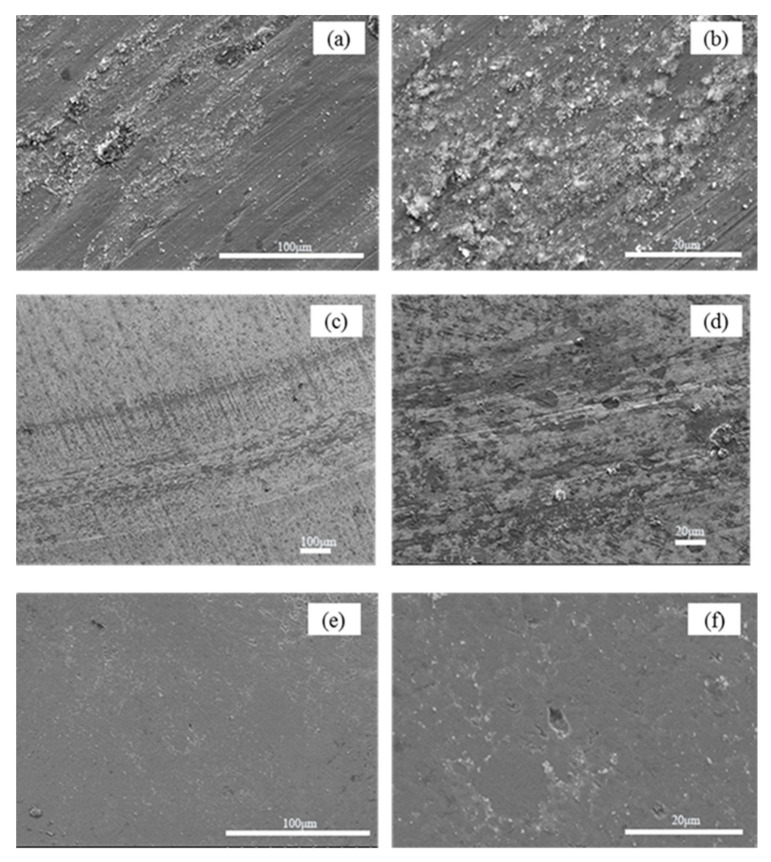
After PoD tests at room temperature, SEM micrographs of wear tracks on the disc surface. (**a**,**b**) H13 hot work tool steel (**c**,**d**) Cemented carbide YG8 (**e**,**f**) CVD coating.

**Figure 10 materials-15-01798-f010:**
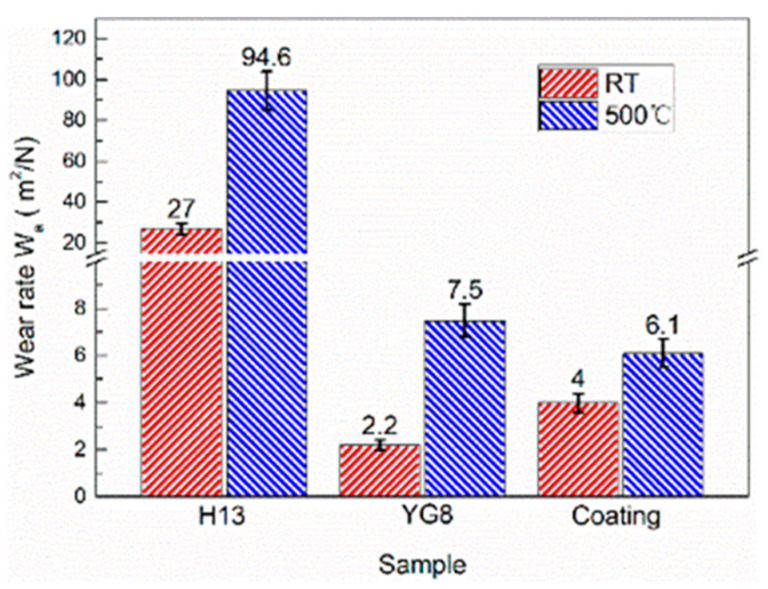
Wear rate of the disks at room temperature and 500 °C. The white line crossing bar of H13 represents the range of the ordinate-axis Y break.

**Figure 11 materials-15-01798-f011:**
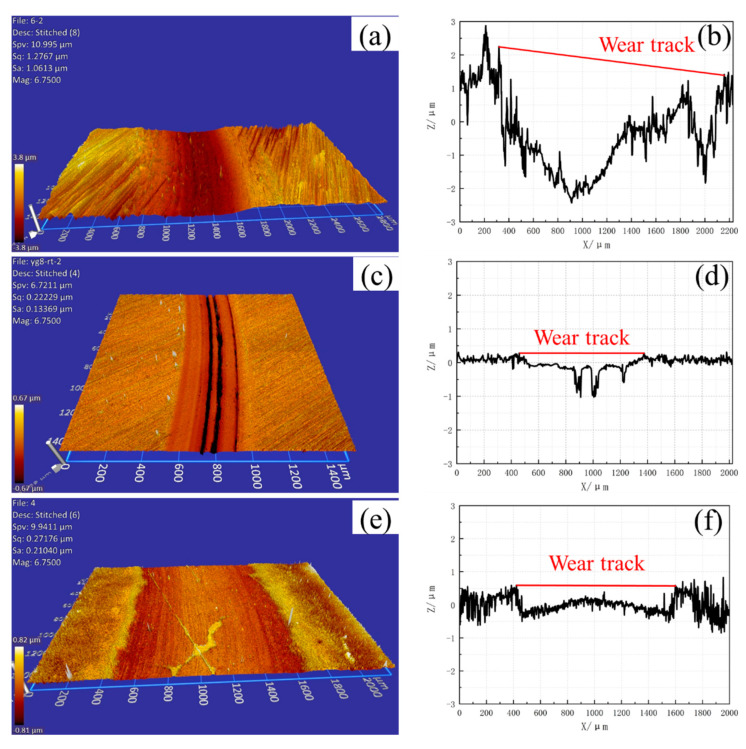
Wear surface after PoD test and wear track profile at room temperature (**a**,**b**) H13 hot work tool steel (**c**,**d**) Cemented carbide YG8 (**e**,**f**) CVD coating.

**Figure 12 materials-15-01798-f012:**
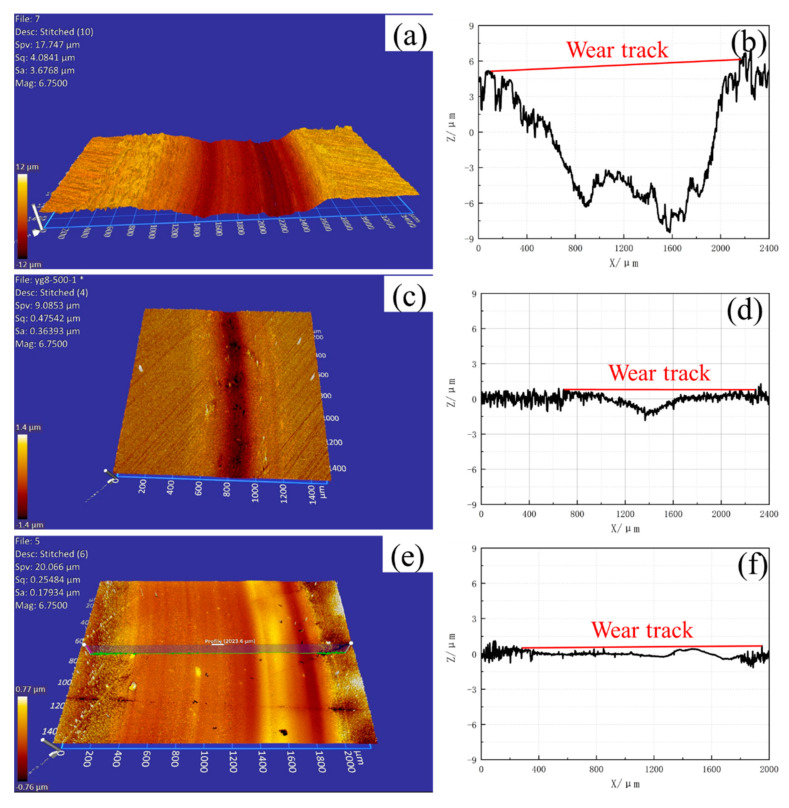
Wear surface after PoD test and wear track profile at 500 °C (**a**,**b**) H13 hot work tool steel, (**c**,**d**) Cemented carbide YG8, (**e**,**f**) CVD coating.

**Figure 13 materials-15-01798-f013:**
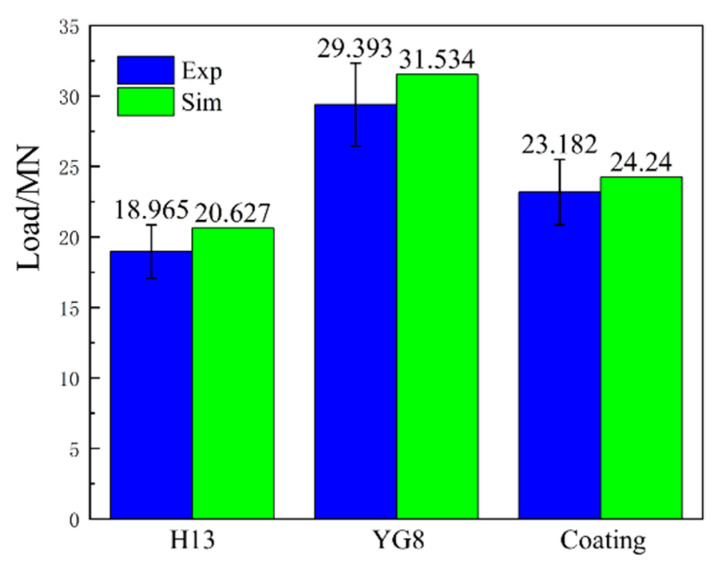
Experimental and predicted Maximum extrusion load.

**Figure 14 materials-15-01798-f014:**
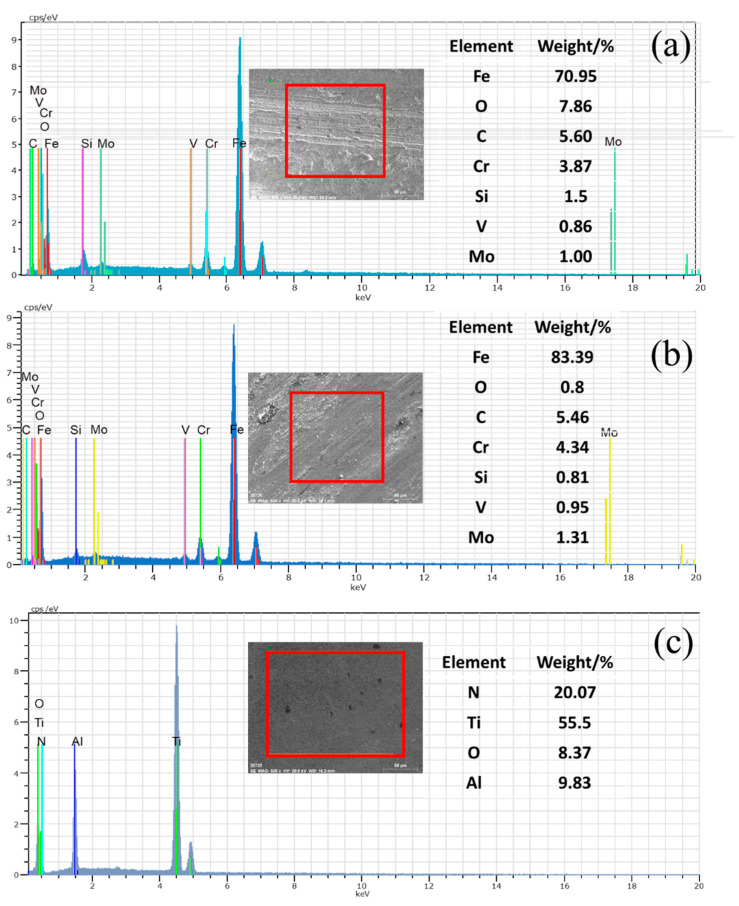
EDS analysis of the wear tracks on the disc surface after PoD tests at room temperature. (**a**) H13 hot work tool steel (**b**) Cemented Carbide YG8 (**c**) CVD Coating.

**Figure 15 materials-15-01798-f015:**
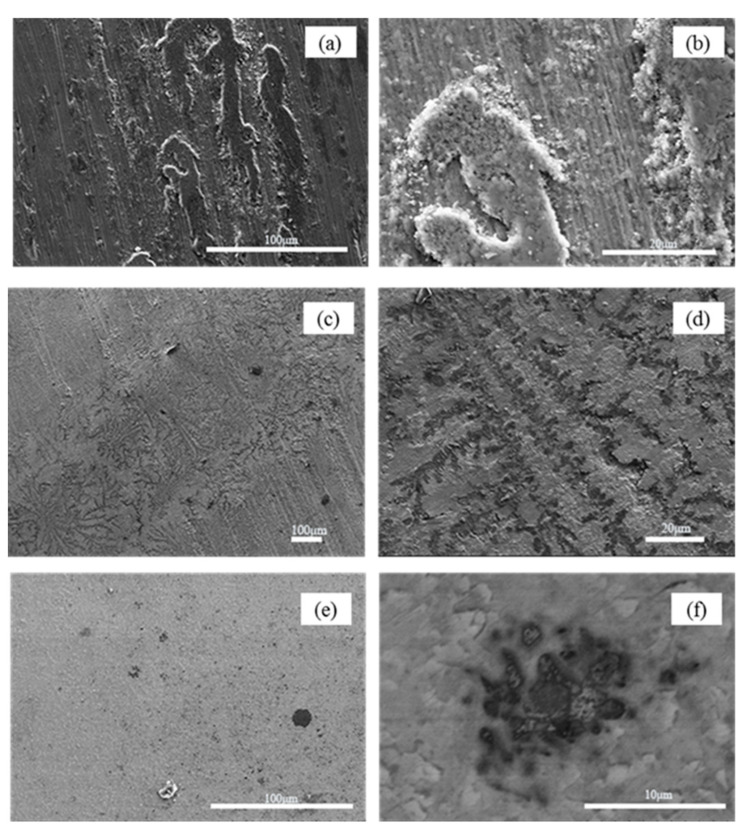
After PoD tests at 500 °C, SEM micrographs of wear tracks on the disc surface. (**a**,**b**) H13 hot work tool steel (**c**,**d**) Cemented carbide YG8 (**e**,**f**) CVD coating.

**Figure 16 materials-15-01798-f016:**
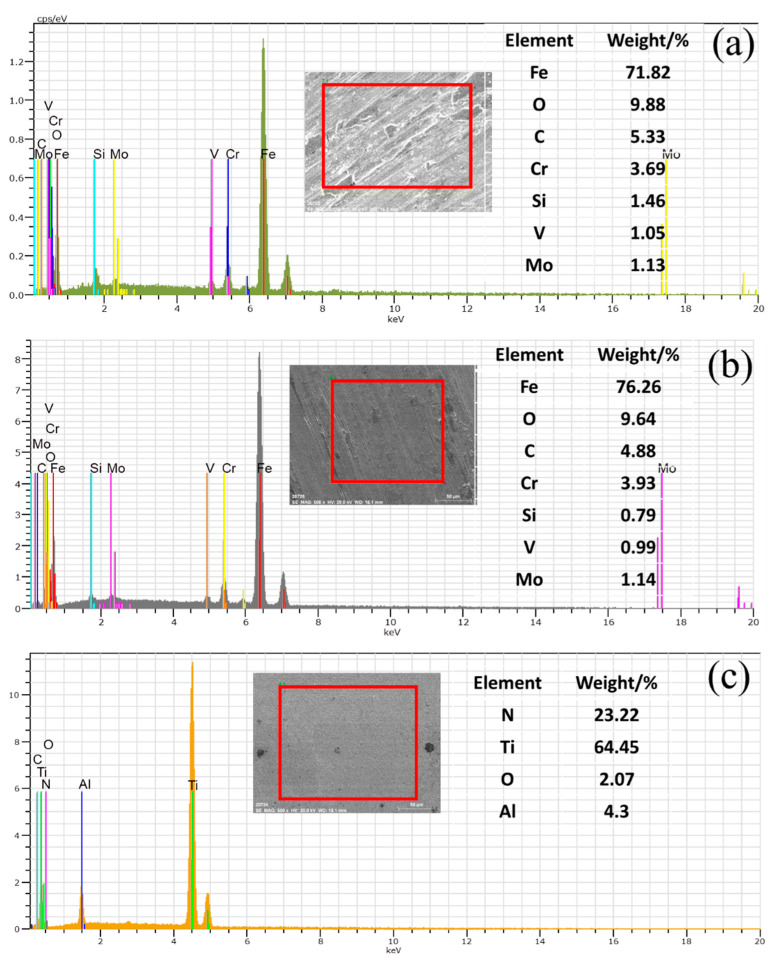
EDS analysis of the wear tracks on the disc surface after PoD tests at 500 °C. (**a**) H13 hot work tool steel (**b**) Cemented carbide YG8 (**c**) CVD coating.

**Table 1 materials-15-01798-t001:** The chemical compositions of the studied die materials and the counter friction specimen.

	WC	Co	C	Si	Cr	Mn	V	Mo	Fe	Others
H13	-	-	0.42	1.04	5.15	0.43	0.9	1.45	bal.	-
YG8	92	8	-	-	-	-	-	-	-	trace
YG6	94	6	-	-	-	-	-	-	-	trace

**Table 2 materials-15-01798-t002:** Results of hardness test.

Material (Type)		Nano Hardness (GPa)	Macro Hardness (HRC)
CVD coating	Surface layer	2.893 ± 0.145	75.9 ± 3.795
	Middle layer	1.688 ± 0.084	
	Lower layer	0.771 ± 0.039	
Cemented Carbide (YG8)		1.529 ± 0.076	70.7 ± 3.535
Cemented Carbide (YG6)		3.896 ± 0.195	87.9 ± 4.395
Hot tool steel (H13)		0.532 ± 0.027	63.78 ± 3.189

**Table 3 materials-15-01798-t003:** The extrusion experiment’s process parameters.

Process Parameters	Value
Diameter of the billet (mm)	228
Length of the billet (mm)	1100
Initial temperature of the billet (°C)	500
Initial temperature of the tool (°C)	480
Extrusion speed (mm/s)	3
Heat convection coefficient (W/m^2^ °C)	30,000

**Table 4 materials-15-01798-t004:** AA3003 and H13 have the following physical properties.

Physical Properties	AA3003	H13 Steel
Density (kg/m^3^)	2635	7870
Thermal conductivity (W/(m K))	180	24.3
Specific heat (J/(kg K))	1090	460
Young’s modulus (Pa)	6.9 × 10^10^	2.1 × 10^11^
Poisson’s ratio	0.3	0.35

## Data Availability

Not applicable.

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
