# Peer review of "Experimental and Numerical Study on Friction and Wear Performance of Hot Extrusion Die Materials"

_materials, 2022, doi:10.3390/ma15051798_

Round 1

Reviewer 1 Report

The authors study mechanical properties and wear resistance of three different extrusion die materials, namely, H13 hot work tool steel, cemented carbide YG8, and chemical vapour deposition (CVD) coating at room temperature and extrusion temperature (500°C). The coefficients of friction and wear rates of the materials are compared. Corresponding wear mechanisms are discussed based on the optical profiler and SEM studies, and corresponding recommendations are drawn. The work is worth to be published because of its high scientific and practical quality. However, the authors should correct some shortcomings to make the manuscript acceptable for publication.

(1) Recently published articles (2018-2021) on the manuscript topic should be added to the reference list.

(2) Instead of “hot tool steel H13 steel” the authors should use the term “H13 tool steel” or “H13 hot work tool steel”.

(3) Please provide the table of chemical compositions of the studied die materials and the counter friction specimen.

(4) The authors should provide marking for cemented carbide materials in accordance with ISO WC-Co grades.

(5) Please use the term “Coefficient of friction” and corresponding abbreviation “COF” instead of “Friction coefficient”.

(6) Line 103: please correct the phrase “sputtering with Arion beams”.

(7) Lines 217-218: the statement for H13 steel should refer to Fig.14a,b, whereas for cemented carbide YG8 it should refer to Fig.14c,d. Please correct this shortcoming.

(8) In Fig.4b, the scale cannot be recognized which does not allow estimating the coating thickness. Besides, no value of this parameter is given in the text.

(9) Lines 286-290: The sentence “In our study, the order of maximum extrusion force of different types of dies obtained from FE simulation is consistent with the order of friction coefficient measured by friction and wear test that the maximum friction coefficient is the cemented carbide YG8, followed by coating die, and the hot tool steel H13 is the smallest as shown in Fig. 9” is unclear. I propose to replace the sentences (lines 278-282) “Fig. 13 illustrates the maximum extrusion load obtained from FE simulation and experimental extrusion. It can be obviously found that the maximum extrusion force of different types of dies is different from the maximum extrusion force of cemented carbide YG8 die is the largest, followed by coating die, and the extrusion force of die steel H13 is the smallest” with the sentences “It is well known that the greater the friction coefficient between the die and the billet, the greater the extrusion force is required. In our study, the relative order of COF values for different die materials measured by friction and wear test is as follows: the cemented carbide YG8 with the maximum COF is followed by CVD coating with a lower COF and H13 hot work tool steel with the smallest one as shown in Fig. 9. The relative order of maximum extrusion forces for different types of dies obtained from FE simulation and experimental extrusion (Fig. 13) is consistent with that of COFs”. The sentences (lines 284-290) “As we all know, the greater the friction coefficient between the die and the billet, the greater the extrusion force is required. In our study, the order of maximum extrusion force of different types of dies obtained from FE simulation is consistent with the order of friction coef-ficient measured by friction and wear test that the maximum friction coefficient is the cemented carbide YG8, followed by coating die, and the hot tool steel H13 is the smallest as shown in Fig. 9” should be removed.

(10) The authors should substantiate how the wear tests of each of the die materials in a couple with a cemented carbide bolt are related to the operation conditions (hot extrusion process), i.e. wear of these materials in a couple with an aluminum alloy. Please provide corresponding references.

(11) In Table 3, please replace “Density (kg m3)” with “Density (kg/m3)”. Young’s modulus for H13 steel should be 2.1e11 Pa (not 2.1e10 Pa). Please correct this shortcoming. If the authors used the value of Young’s modulus for H13 steel 2.1e10 Pa, they should substantiate whether the FE calculation results presented in the work are correct or not.

(12) In Fig.15 and Fig.17, each figure caption contains six positions (letters from “a” to “f”) whereas in the figure body there are only three positions (letters from “a” to “c”). Please correct these shortcomings.

Reviewer 2 Report

The manuscript addresses the friction and wear behavior of hot tool steel, cemented carbide, and a CVD coating as aluminium alloy hot extrusion die materials. To this end, macro- and nano-indentation hardness measurements as well as pin-on-disk tribo-tests were carried out at room temperature and at 500 °C. The wear mechanisms were analyzed by optical profilometry and SEM. The frictional results were used as inputs for a FEM simulation of an extrusion process.

It was reported that hot tool steel featured the lowest hardness, the highest wear and high friction at RT. At 500°C, however, friction was the lowest, also leading to the lowest extrusion loads in an FEM simulation and the respective experiments. The cemented carbide showed the lowest friction and wear at RT while the CVD coating had the highest wear resistance at elevated temperature. The wear mechanisms of hot tool steel, cemented carbide, and CVD coatings at 500℃ were adhesion wear, abrasive wear, and fatigue wear, respectively.

The overall topic is of interest for the research community and suitable for publication in the journal Materials. The methods and presented results & discussion are scientifically sound. But I have a couple of comments that should be addresses prior to consideration for publication:

  1. The abstract should be written more concise and include more information on the findings and not just a description of what was done.
  2. The keywords of the paper should be more specific. There is no need to repeat elements from the title.
  3. The first paragraph of the introduction should be substantiated with references. I don't think Fig. 1 and 2 are necessary.
  4. Line 42: What is meant by "frictional stress on bearing surfaces"? This is misleading. 
  5. This is a subjective comment: I prefer when references to Figures or Tables are formulated passively ("something is shown in Figure 1" instead of "Figure 1 shows something"), because the authors show something with the item and the Figure/Table itself cannot perform any active actions. The authors can feel free to consider this or not.
  6. Fig. 3 can be omitted.
  7. Line 99: It should be "∅5 mm" instead of "Φ".
  8. While there are the sub-sections 3.1 and 3.2, there is no section 3.
  9. Why was a cemented carbide pin used as counter-body? To what extent does this fit to the application? Wouldn't aluminium make sense to study an aluminium extrusion process? I doubt that the wear mechanisms are transferable then.
  10. Fig. 4b is hard to read (red font on dark background). The measurements in Fig. 4 should be shown and discussed in the results section. Section 2 is for materials and methods only!
  11. More information on the hardness measurements should be provided (measurement parameters, type of indenter, indentation depths, force, times, number or measurements, distance betweem measurements etc.). For the results, standard deviations should be provided.
  12. Section 2.3: The pressure should be given in MPa and the velocity in m/s (like the typical values provided in the introduction).
  13. Fig. 6: Red font on multi-clored background is hard to read.
  14. Figures should be shown after their first mention in the text. Section 4.1 refers to Fig. 8 (which is after section 3) and Fig. 14 (which is in section 4.4). This makes the reader scroll a lot through the manuscript. I suggest the authors to re-structure the paper into a results section where they can just show and describe the results and then add another discussion section wear they discuss and connect the results. This might be the better option here.
  15. Line 215-216 reads confusing, please re-write.
  16. The wear rate was always higher at elevated temperature while this was not always the case for friction. Please discuss on that.
  17. It should be clarified what is shown with the scatter bars in Fig. 9 and 12. Is it the the minimum and maximum values, the standard deviation or the propagated standard deviation. The latter would be desirable.
  18. Although it appears clear, it might be mentioned that the 500 °C COFs were used as simulation inputs. Also, a sentence that compares the qualitative differences between Fig. 13 and the COFs in Fig. 9 (blue) should be added.
  19. Fig. 13: Were the experimental measurements conducted in replicates? Is it possible to add scatter bars?

Round 2

Reviewer 2 Report

The paper has been satisfactorily revised and I believe it can now be accepted in its present form.